# Response of Different Treatment Protocols to Treat Chronic Non-Bacterial Osteomyelitis (CNO) of the Mandible in Adult Patients: A Systematic Review

**DOI:** 10.3390/ijerph17051737

**Published:** 2020-03-06

**Authors:** Maximilian Timme, Lauren Bohner, Sebastian Huss, Johannes Kleinheinz, Marcel Hanisch

**Affiliations:** 1Department of Cranio-Maxillofacial Surgery, University Hospital Münster, Albert-Schweitzer-Campus 1, Building W 30, D-48149 Münster, Germany; Maximilian.timme@ukmuenster.de (M.T.); lauren.bohner@ukmuenster.de (L.B.); johannes.kleinheinz@ukmuenster.de (J.K.); 2Department of Pathology, University Hospital Münster, Germany, Domagkstrasse 17, D-48149 Münster, Germany; sebastian.huss@ukmuenster.de

**Keywords:** osteomyelitis, mandible, CNO, non-suppurative osteomyelitis, chronic non-bacterial osteomyelitis, SAPHO, diffuse sclerosing osteomyelitis

## Abstract

(1) Background: Chronic non-bacterial osteomyelitis (CNO) is an autoinflammatory bone disease of finally unknown etiology, which can occur alone or related with syndromes (chronic recurrent multifocal osteomyelitis—CRMO; synovitis, acne, pustulosis, hyperostosis and osteitis syndrome—SAPHO). The involvement of the mandible is rather rare. (2) Methods: We carried out a systematic literature search on CNO with mandibular involvement, according to the “Preferred Reporting Items for Systematic Reviews and Meta-Analyses” (PRISMA) guidelines, considering the different synonyms for CNO, with a special focus on therapy. (3) Results: Finally, only four studies could be included. A total of 36 patients were treated in these studies—therefore, at most, only tendencies could be identified. The therapy in the included works was inconsistent. Various therapies could alleviate the symptoms of the disease. A complete remission could only rarely be observed and is also to be viewed against the background of the fluctuating character of the disease. The success of one-off interventions is unlikely overall, and the need for long-term therapies seems to be indicated. Non-steroidal anti-inflammatory drugs (NSAIDs) were not part of any effective therapy. Surgical therapy should not be the first choice. (4) Conclusions: In summary, no evidence-based therapy recommendation can be given today. For the future, systematic clinical trials on therapy for CNO are desirable.

## 1. Introduction 

Chronic non-bacterial osteomyelitis (CNO) is an autoinflammatory disease of unclear etiology. The association of CNO with various diseases, including inflammatory, rheumatic, dermatological, and inflammatory bowel diseases (IBD), is reported [1,2,3,4,5,6]. Cytokine imbalance appears to be the decisive factor in CNO pathogenesis. Patients with CNO show an underproduction of anti-inflammatory cytokines (interleukin-10) and overproduction of proinflammatory cytokines (tumor necrosis factor-α (TNFα) and interleukin-1). Due to the spontaneous hyperproduction of proinflammatory cytokines without autoimmunity features, CNO is classified as an autoinflammatory disease, thus falling into the rheumatic group of diseases [7,8,9,10,11,12,13,14]. The oral and maxillofacial surgery literature also discusses further factors as etiology of CNO. These range from bacterial infections of low virulence to functional causes. However, it must be considered that these causes are not considered in the scientific literature on the rheumatology of CNO in regard to other localizations [15,16,17,18].

Patients regularly present with a very painful swelling along the mandible, often associated with trismus [19]. The initial patient consultation with regard to the CNO of the mandible often takes place at the dentist since the pain is misinterpreted as toothache or, in combination with swelling, as an abscessing process. A possible moderate increase in CRP (C-reactive protein) in the serum within the scope of CNO may also lead to misinterpretation [14].

To date, there are no specific guidelines for the treatment of chronic non-bacterial osteomyelitis in adult patients. The therapeutic strategies described in the literature range from conservative functional therapy and drug therapy with non-steroidal anti-inflammatory drugs (NSAIDs) to complete resection of the mandible with plastic reconstruction [4,17,20]. Since the disease is rarely diagnosed in adult patients, randomized clinical trials are scarce. Nonetheless, observational studies show the response of different treatment protocols with a long-term follow-up. In this paper, the current available literature regarding treatment protocols for CNO was systematically updated. 

## 2. Methods

The present review was conducted in accordance with the guidelines available at the “Preferred Reporting Items for Systematic Reviews and Meta-Analyses” (PRISMA) [21] and was approved by the ethics committees of the Chamber of Physicians of Westfalen-Lippe (Ärztekammer Westfalen-Lippe) and the University of Münster in Germany (Ref. No. 2019–232-f-N).

The focused question of this review was: what is the response of different treatment protocols to treat non-suppurative osteomyelitis of the jaw in adult patients? PICOS was defined as the following: P = patients above 18 years diagnosed with non-suppurative osteomyelitis/chronic non-bacterial osteomyelitis (CNO) of the jaw; I = treatment protocol; C -; O = disappearance/persistence of clinical and radiological symptoms; S = clinical studies. 

### 2.1. Eligibility Criteria

The inclusion criteria consisted of clinical studies describing the treatment protocol and follow-up of adult patients diagnosed with CNO of the jaw. Osteomyelitis as a manifestation of a primary disease/syndrome (SAPHO) was also considered for inclusion.

Conversely, exclusion criteria comprised the following: A—articles that were not written in English; B—case reports, letters, conference abstracts and literature reviews; C—case series or clinical studies with less than 5 patients; D—patients under 18 years; E—studies not describing the treatment protocol or the outcome of the disease.

### 2.2. Information Sources

The electronic search was conducted on the databases Pubmed (MedLine), Scopus, Cochrane and Embase from 5 March to 29 May, 2019. A main search strategy based on the PICOS terms was developed for this purpose (Table 1) and applied on Pubmed (Medline). After, the main search was modified to be used according the requirements of each database.

### 2.3. Study Selection

The study selection was performed independently by two reviewers (MT and LB). In case of disagreement, both authors discussed with the third reviewer (MH) until they achieved a mutual consensus. 

In the first phase, study selection was performed based on the screening of articles by reading the title abstracts. After, screened articles which responded to the inclusion criteria were selected to be read in full. Articles were then excluded according to the requirements of this review, and only those which were considered relevant were included for analysis. The management of studies was performed using the software Rayyan (Qatar Computer Research Institute, Ar-Ryyan, Qatar). 

### 2.4. Data Collection Process and Items

All relevant information regarding study design, treatment protocol, and outcome was extracted from the included articles by both reviewers. If relevant information was missing, the first author was contacted by email, and a non-response from the author resulted in the exclusion of the article.

### 2.5. Risk of Bias within Studies

Observational studies were assessed by the tool “Methodological Index for Non-Randomized Studies” (MINORS), which consists of 7 items regarding the methodological quality of non-comparative studies. Items are scored as the following: 0 = not reported; 1 = reported inadequately; 2 = reported adequately [22].

## 3. Results 

### 3.1. Study Selection

After removing duplicates, 1523 studies were screened from the electronic databases. After reading titles and abstracts, 59 papers were selected for full-text reading and, from these, 55 studies were excluded according to the exclusion criteria. Thus, only four studies were included for analysis (Figure 1).

### 3.2. Study Characteristics

Three of the included studies were retrospective studies, and one of them was a prospective study (Table 2 and Table 3). 

In total, 36 patients aged 19–78 years were treated with different therapeutic approaches. The most common symptoms were pain, trismus and swelling. Treatment protocol varied from surgical intervention to short- or long-term antibiotic therapy, such as prescription of non-steroidal anti-inflammatory medication, corticosteroids and bisphosphonates. The follow-up period ranged from 12 months to 19 years. In general, all authors reported a decrease in clinical or radiological symptoms, although complete cessation was not seen in all cases. Long-term antibiotic therapy was completely successful in 77.8% of the cases [19], whereas bisphosphonate therapy was completely effective against pain in 28.57% of the treated patients [24].

Jacobsson and Hollender (1980) evaluated 11 patients above 18 years of age who were diagnosed with diffuse sclerosing osteomyelitis (DSO), one of the common synonyms of CNO, during a period of 3–19 years. Patients were treated with short- and long-term antibiotic therapy and when clinical and radiological symptoms were more evident. Additionally, non-steroidal and steroidal analgesics were also prescribed in cases of pain. The authors reported that long-term antibiotic therapy decreased the occurrence of exacerbation, whereas decortication resulted in a relief of pain during a period of six months to one year [23].

Kujipers et al. (2011) [24] reported six clinical cases of adults (ranging 23–78 years) diagnosed with DSO. Patients were treated previously with different therapies, but they were not effective. Thus, patients were treated with intravenous pamidronate 15 mg and followed up during a period of 18–46 months. The pain was completely eliminated for only one patient. Nonetheless, the symptoms were reduced for all patients, and the disease activity shown in the Tc-scan was decreased.

Montonen et al. (2001) [25] investigated pain relief using a VAS (visual analogue scale) scale after the administration of intravenous disodium clodronate in patients diagnosed with DSO. Patients were treated with the medicament intravenously and compared to a placebo group. They were followed up during 1 week and 1, 3, 6 and 12 months after treatment. After six months, the treatment group presented with greater pain relief in comparison to the placebo group.

Yoshii et al. [19] assessed the efficacy of a long-term antibiotic therapy for the treatment of DSO. Nine patients with intermittent pain, swelling and trismus were included. Treatment was conducted with roxythromycin 300 mg, and the course of the treatment was determined according to the change in symptoms. Thus, the treatment duration ranged between 68 days and 66 months. In total, seven patients were reported to show good clinical efficacy.

### 3.3. Risk of Bias within Studies

Most of the included studies exhibited low quality when reporting their results. All studies showed a deficiency regarding statistical analysis issues and endpoints related to the aim of the study. Additionally, three studies showed a weakness in reporting items related to the collection and assessment of data, such as an inappropriate follow-up period considering the aim of the study. The follow-up period was considered inadequate for three of four studies (Table 4). 

## 4. Discussion

The present study aims to provide an overview of different treatment protocols available for CNO in adult patients. The age range in the studies included was from 19 to 78 years. This represents nearly the total time span of adult life. In principle, the included collective can be used to make conclusions about CNO in adult patients. Only one of the included studies was a prospective, randomized and double-blinded trial [25]. Consequently, this study also achieved the highest MINORS score [22]. The treatment approaches within the included studies were very heterogeneous, whereby in two cases, the substance group of bisphosphonates was used [24,25].

One problem with the evaluation is that there is no clear definition of therapeutic success in the various data included. However, it can already be concluded that a definitive cure can only be achieved in the rarest of cases, since otherwise, this would be the standard norm for successful treatment.

### 4.1. Follow-Up

The very different follow-up periods from 12 months to 19 years might have a negative effect on the reliability of the results. In view of the fluctuating character of the disease, a long-term follow-up period is desirable. In particular, since Jacobsson et al. describe recurrences after 6 to 12 months. In the studies included, the follow-up of Yoshii et al. [19] was only 12 months. The treatment duration in their study, on the other hand, was very long at up to 66 months. Here, it must be noted that reliable data on recurrences are of course also missing, which makes even statements on the best follow-up period difficult.

### 4.2. Surgical Therapy

The current literature overall describes surgical procedures ranging from decortications to partial resections of the mandible [4,17,26,27,28,29]. In the present data included, only Jacobsson and Hollender reported a surgical procedure as part of the treatment protocol. However, a frequent return of symptoms after 6–12 months following decortication was described by the authors, since permanent pain relief could only be achieved in one patient after decortication [23]. In this sense, it may be assumed that decortication does not prove to be beneficial when compared with non-surgical approaches.

Furthermore, the concomitant morbidity of surgical procedures, in particular of partial resections, should be considered. In a case report known from the literature, even the transplanted fibular interponate after partial resection was affected by the disease, and healing could not be guaranteed even after radical surgical resection [30]. 

### 4.3. Pharmacological Therapy

This brings pharmacological therapy for CNO into focus, which can also be regarded as consensusable today against the background of other reviews [7,20]. The literature also contains a large number of case reports and case series on conservative therapy for CNO. However, these could not be considered for the present study since they do not allow any real conclusions to be drawn about the evidence. 

### 4.4. Antibiotics

From the work included, Yoshii et al. have described the successful use of antibiotics for CNO therapy, and Jacobsson and Hollender also report positive effects of antibiotic therapy, at least in the short term. This is difficult to reconcile with the assumed pathomechanism of the disease. Possibly, the positive effect of antibiotics observed by some authors is related to the fact that in patients with rheumatic disorders, antibodies of anaerobic microorganisms, especially periodontal bacteria, may be increased [31]. However, the approach that seems to be more appropriate is that some antibiotics, especially those that are repeatedly discussed for the treatment of rheumatic diseases, have anti-inflammatory partial effects that run independently of their anti-infective effects [32]. Overall, the potency of roxythromycin in CNO as described by Yoshii et al. could thus be attributed to its anti-inflammatory effects [33,34]. Hence, the results should in no case lead to the conclusion that there is an indication for a general antibiotic therapy for CNO. Unfortunately, it cannot be deduced from the available data to what extent therapy with roxythromycin is superior to cortisone monotherapy. 

### 4.5. Antiresorptive Drugs

Studies on CNO therapy with antiresorptive drugs are now also available. The results of the literature analyzed here at least provide indications of their efficacy [24,25]. For this reason, Kuijpers et al. [24] reported a decrease in symptoms and a reduced need for pain medication; one patient was symptom-free after a single dose of pamidronate. Monotonen et al. [25] also reported a significant decrease in pain symptoms after i.v. application of disodium clodronate after six months compared to the control group, which was treated with placebo. However, healing does not seem to be possible with a single dose. No statement can be made today on the best therapy for a recurrence. 

The observations on the potential effects of bisphosphonates are also consistent with other case reports in the literature [5,35,36]. 

In theory, long-term bisphosphonate therapy would be conceivable—albeit potential side effects would then have to be weighed up [37].

The effectiveness of antiresorptives can also be theoretically reconstructed via the effect of bisphosphonates on the TNF-alpha receptor family [36,38]. In addition, first data for the use of denosumab show promising results [39,40]. Unfortunately, however, the reports could not be considered for this paper. Therefore, concrete recommendations cannot be given.

### 4.6. Use of NSAIDs

Although, especially for children, the use of NSAIDs is suggested frequently, the response rates can be rated as low [16,41]. In the studies considered for this article, NSAIDs are mentioned as prior medication. It can therefore be concluded that the pre-treatment was insufficient [24,25]. The individual effect of NSAIDs cannot be assessed as a whole due to the limited data available. However, contrary to the assessments of other current reviews, the authors recommend not to overestimate the effect of NSAIDs and not to delay other therapies by attempting a therapy using NSAIDs [20]. Considering the comparatively low adverse drug effects, however, NSAIDs can be regarded as concomitant therapy [20,42].

### 4.7. Nomenclature Recommendation

The term primary chronic osteomyelitis (PCO) comes from the Zurich Classification [1]. Although PCO is defined there as “a non-pusturous, non-fistuating and non-sequestrating, chronic-inflammative form of osteomyelitis with unknown aetiology,” the purely conceptual distinction from bacterial osteomyelitis in this classification is vague, especially since it is still distinguished from secondary chronic osteomyelitis, which in turn is a bacterial form [1]. 

### 4.8. Limitations of the Review 

The authors chose to perform this review systematically in order to assess the clinical and scientific relevance of the existent literature. However, the scarcity of literature regarding this theme was a limitation when responding to the focused question of this review. Randomized clinical trials are desirable for the future. An inconsistent nomenclature of CNO was observed; however, the authors used the term chronic non-bacterial osteomyelitis (CNO) because of its better delimitability.

## 5. Conclusions 

In general, the described treatment protocols brought only temporary relief of symptoms. Recurrences were frequent, which reflects the chronic character of the disease. Due to the low number of included studies, no conclusion could be drawn regarding the efficacy of the reported treatment protocols.

## Figures and Tables

**Figure 1 ijerph-17-01737-f001:**
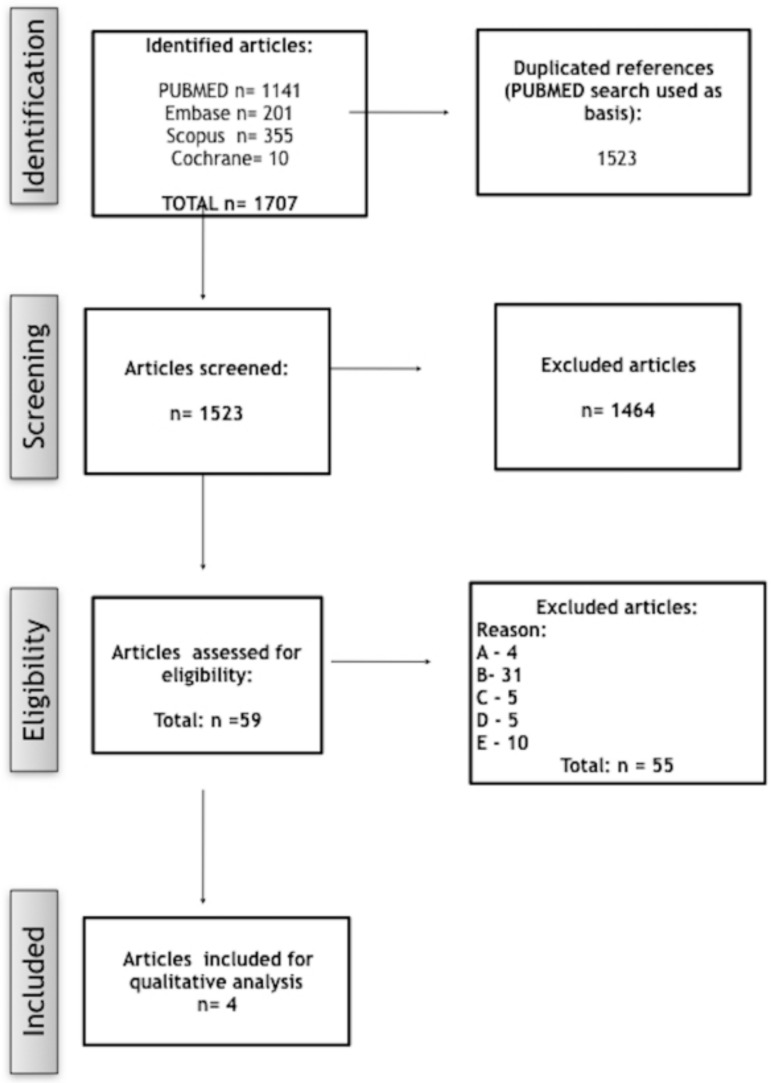
Fluxogram showing the search strategy.

**Table 1 ijerph-17-01737-t001:** Search strategy on PubMed (Medline).

P	#1	((mandible) OR mandibular) OR jaw	180,180
I	#2	(((((((((osteomyelitis [MeSH Terms]) OR "diffuse sclerosing osteomyelitis") OR "chronic non-bacterial osteomyelitis") OR CNO) OR DSO) OR PCO) OR SAPHO) OR CRMO))	28,673
O	#3	((((treatment) OR therapy) OR follow-up) OR outcome)	10,715,195
P+I+O	#1+ #2+ #3	((((((((((((osteomyelitis [MeSH Terms]) OR "diffuse sclerosing osteomyelitis") OR "chronic non-bacterial osteomyelitis") OR CNO) OR DSO) OR PCO) OR SAPHO) OR CRMO)))) AND (((mandible) OR mandibular) OR jaw)) AND ((((treatment) OR therapy) OR follow-up) OR outcome)	1141

**Table 2 ijerph-17-01737-t002:** Data summary of the included studies.

Authors	Country	Study design	Patients (n)	Term	Gender	Age	Symptoms
Jacobsson and Hollender. [23]	Sweden	Retrospective study	11	DSO	F (7), M (4)	22–53	Pain, swelling, trismus, ankylosis from TMJ
Kuijpers et al. [24]	Netherlands	Retrospective study	6	DSO	F (5), M (1)	23–78	
Montonen et al. [25]	Finland	Prospective, randomized, double-blind and placebo controlled.	10	DSO	F (8), M (2)	31–77	Pain
Yoshii et al. [19]	Japan	Retrospective study	9	DSOM	F (5), M (4)	19–70	Pain, swelling, trismus

**Table 3 ijerph-17-01737-t003:** Treatment protocol of the included studies.

Authors	Imaging	Previous Treatment	Treatment	Therapeutic Regime	Outcome	Follow-Up
Jacobsson and Hollender. [23]	Radiographic and scintigraphic examinations	Short- and long-term antibiotic therapy, decortication, tooth extraction	Decortication (7), excision, resection, antibiotics, cortisone	Predinisolone 20/5 mg (12 d), penicillin (3 months)	Long-term antibiotic therapy decreased the interval between exacerbations and pain/decortication ceased the symptoms for 6–12 months	3–19 y
Kuijpers et al. [24]	Panoramic radiograph, CT	Analgesics (acetaminophen, NSAIDs), antibiotics (e.g., doxycycline, amoxicillin, vibramycin, clindamycin), physiotherapy, corticosteroids and/or surgery	Bisphosphonate (intravenous dose of 15 mg pamidronate, 3–5 d)	-	Symptoms decreased or disappeared	18–46 months
Montonen et al. [25]	Radiological examinations, scintigraphy and orthopantomography	Conservative or surgical therapy	Bisphosphonate	Disodiumclodronate (300 to 900 mg) or placebo intravenously	There was a greater pain relief after 6 em of treatment with bisphosphonate	12 months
Yoshii et al. [19]	Radiographic imaging	Surgical treatment	Long-term roxithromycin	300 mg, oral, daily, 68 d–66 months	Pain was reduced and ceased completely in 7 of 9 patients	12 months

**Table 4 ijerph-17-01737-t004:** Risk of bias within studies MINORS [22].

Risk of bias assessment	Jacobsson et al.	Kuijpers et al.	Montonen et al.	Yoshii et al.	Total
A clearly stated aim	1	0	2	1	4
Inclusion of consecutive patients	1	1	2	2	6
Prospective collection of data	0	1	2	0	3
Endpoints appropriate to the aim of the study	1	1	1	1	4
Unbiased assessment of the study endpoint	0	0	2	1	3
Follow-up period appropriate to the aim of the study	1	1	2	1	5
Loss to follow-up less than 5%	2	1	2	2	7
Prospective calculation of the study size	0	0	0	0	0

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
