# Peer review of "Response of Different Treatment Protocols to Treat Chronic Non-Bacterial Osteomyelitis (CNO) of the Mandible in Adult Patients: A Systematic Review"

_ijerph, 2020, doi:10.3390/ijerph17051737_

Round 1
Reviewer 1 Report
In this study, the authors summarized the current treatment protocols of treating chronic non-bacterial osteomyelitis (CNO) of mandible in adult patients in the literature. The authors emphasized this is a systematic review. However, the reviewer has a concern about the cases used in this review. The authors only included 4 studies related to treatment protocols of CNO, whether or these 4 studies could form an adequate pool of cases to make this review systematic? Other than this concern, the table in Page 5 may need a re-organized and be correctly shown in the manuscript.
Author Response
Reviewer´s response
Dear Reviewers,
We would like to thank you for your effort in reviewing our manuscript. We agree with your suggestions and we tried to improve the manuscript accordingly. All changes are highlighted in red and explained below:
Reviewer 1:
In this study, the authors summarized the current treatment protocols of treating chronic non-bacterial osteomyelitis (CNO) of mandible in adult patients in the literature. The authors emphasized this is a systematic review. However, the reviewer has a concern about the cases used in this review. The authors only included 4 studies related to treatment protocols of CNO, whether or these 4 studies could form an adequate pool of cases to make this review systematic? Other than this concern, the table in Page 5 may need a re-organized and be correctly shown in the manuscript.
The authors decided to conduct this systematic review according to the PRISMA Guidelines in order to standardize the search strategy and studies selection. This strategy was considered relevant to address all studies available about this issue, especially considering the diversity founded on the terminology. The fact that only four observational studies were included was considered a limitation of the study. This was reported at the Discussion topic (lines 281-285):
“4.8 Limitations of the review
The authors chose to perform this review systematically in order to assess the clinical and scientific relevance of the existent literature. However, the scarcity on literature regarding this theme was a limitation to respond the focused question of this review. Randomized clinical trials would be desirable in the future. An inconsistent nomenclature of the CNO was observed, however, the authors used the term chronic non-bacterial osteomyelitis (CNO) because of its better delimitability.”
Furthermore, the table 5 was reorganized.
Reviewer 2:
The paper is very interesting, clear and well written. There is a logical flow to the topics. The evidence comes from a wide variety of valid sources. Conclusion follows clearly from the arguments presented. The bibliography is complete and reflects appropriate sources.
In my opinion, the paper is well written and with logical flow of data. As an author of papers on rare diseases, I believe that because chronic non-bacterial osteomyelitis are a rare non-infectious inflammatory bone disease, in literature there are not many epidemiological studies but above all case reports and case series which clearly are excluded from meta-analyzes. Unfortunately this information does not emerge from the data analysis. Therefore if an annotation can be done, I believe, this aspect should be clarified and underlined in the discussion. Furthermore, I would recommend to improve the conclusion in order to define the scientific contribution and to make a significance of the study.
We rephrased and improved the discussion section as well as the conclusion section.
Reviewer 3:
In this systematic review, the authors investigated the therapeutic patterns and responses of treatment to chronic non-bacterial osteomyelitis (CNO) based on various symptoms. Various investigations have been carried out well so that the problems of current diagnosis or treatment can be pinpointed clearly. However, the authors argue in advance of their negative views on the known diagnosis and treatment approaches for CNO (line 40-44). Because of this paragraph, the questions would be raised about the overall usefulness of this review. In addition, the paragraphs described in lines 61 through 69 points out the diversity of existing diagnostic methods and suggest that there is no appropriate recommendation. So this reviewer wonders whether the authors' purpose is only to criticize existing diagnosis and therapeutics. If that is not the case, it will be needed to clarify what problems the authors are trying to improve the concerns about the diagnosis and treatment of CNO. Finally, a discussion of the clinically resolvable options must be made following raised problems in each diagnosis or treatment.
The purpose of this article was not to report a critical view regarding the current treatment and diagnostics. Thus, the cited paragraphs (lines 40-44 and lines 61-69) were deleted. The rationale for this review was reformulated at the Introduction (lines 62-68) as described below:
“To date, there are no specific guidelines for the treatment of chronic non-bacterial osteomyelitis. The therapeutic strategies described in the literature range from conservative functional therapy and drug therapy with NSAIDs (Non-steroidal anti-inflammatory drugs) to complete resection of the mandible with plastic reconstruction [4,17,20]. Since the disease is rarely diagnosed in adult patients, randomized clinical trials are scarce. Nonetheless, observational studies show the response of different treatment protocols with a long-term follow-up. In this paper, the current available literature regarding the treatment protocols of CNO was systematically updated.”
In addition, the purpose of the study was cited on Discussion (line 192-193), as described below:
This statement was deleted:
“The aim of the present article was to derive a therapy for the CNO of the mandible that was evidence-based and effective. This goal could not be achieved in view of the studies finally included. All in all, only the data of 36 patients could be considered, of which 25 were women and 11 men. Many CNO data had to be disregarded because they are presented in case reports or smaller case series.”
This statement was included:
“The present study aimed to provide an overview of different treatment protocols available for CNO in adult patients.”
In this regards, the conclusion was also changed (lines 288-291):
“In general, the described treatment protocols brought only temporary relief of symptoms. Recurrences were frequent, which reflects the chronic character of the disease. Due to the low number of included studies, no conclusion could be drawn regarding the efficacy of the reported treatment protocols.”
Reviewer 2 Report
The paper is very interesting, clear and well written. There is a logical flow to the topics.The evidence comes from a wide variety of valid sources. Conclusion follows clearly from the arguments presented. The bibliography is complete and reflects appropriate sources.
In my opinion, the paper is well written and with logical flow of data. As an author of papers on rare diseases, I believe that because chronic non-bacterial osteomyelitis are a rare non-infectious inflammatory bone disease, in literature there are not many epidemiological studies but above all case reports and case series which clearly are excluded from meta-analyzes. Unfortunately this information does not emerge from the data analysis. Therefore if an annotation can be done, I believe, this aspect should be clarified and underlined in the discussion. Furthermore, I would recommend to improve the conclusion in order to define the scientific contribution and to make a significance of the study.
Author Response

(The authors gave the same response as above.)

Reviewer 3 Report
In this systematic review, the authors investigated the therapeutic patterns and responses of treatment to chronic non-bacterial osteomyelitis (CNO) based on various symptoms. Various investigations have been carried out well so that the problems of current diagnosis or treatment can be pinpointed clearly. However, the authors argue in advance of their negative views on the known diagnosis and treatment approaches for CNO (line 40-44). Because of this paragraph, the questions would be raised about the overall usefulness of this review. In addition, the paragraphs described in lines 61 through 69 points out the diversity of existing diagnostic methods and suggest that there is no appropriate recommendation. So this reviewer wonders whether the authors' purpose is only to criticize existing diagnosis and therapeutics. If that is not the case, it will be needed to clarify what problems the authors are trying to improve the concerns about the diagnosis and treatment of CNO. Finally, a discussion of the clinically resolvable options must be made following raised problems in each diagnosis or treatment.
Author Response

(The authors gave the same response as above.)
